# OBJECT-CENTRIC LEARNING WITH SLOT MIXTURE MODELS

## ABSTRACT

Object-centric architectures usually apply some differentiable module on the whole feature map to decompose it into sets of entity representations called slots. Some of these methods structurally resemble clustering algorithms, where the center of the cluster in latent space serves as a slot representation. Slot Attention is an example of such a method as a learnable analog of the soft K-Means algorithm. In our work, we use the learnable clustering method based on Gaussian Mixture Model, unlike other approaches we represent slots not only as centers of clusters but we also use information about the distance between clusters and assigned vectors, which leads to more expressive slot representations. Our experiments demonstrate that using this approach instead of Slot Attention improves performance in different object-centric scenarios, achieving the state-of-the-art performance in the set property prediction task.

## 1 INTRODUCTION

In recent years, interest in object-centric representations has greatly increased (Greff et al. (2019); Burgess et al. (2019); Li et al. (2020); Engelcke et al. (2020; 2021); Locatello et al. (2020)). Such representations have the potential to improve the generalization ability of machine learning methods in many domains, such as reinforcement learning (Keramati et al. (2018); Watters et al. (2019a); Kulkarni et al. (2019); Berner et al. (2019); Sun et al. (2019)), scene representation and generation (El-Nouby et al. (2019); Matsumori et al. (2021); Kulkarni et al. (2019)), reasoning (Yang et al. (2020)), object-centric visual tasks (Groth et al. (2018a); Yi et al. (2020); Singh et al. (2021b)), and planning (Migimatsu & Bohg (2020)).

Automatic segmentation of objects on the scene and the formation of a structured latent space can be carried out in various ways (Greff et al. (2020)): augmentation of features with grouping information, the use of a tensor product representation, engaging ideas of attractor dynamics, etc. However, the most effective method for learning object-centric representations is Slot Attention Locatello et al. (2020). Slot Attention maps the input feature vector received from the convolutional encoder to a fixed number of output feature vectors, which are called Slots. As a result of training, each object is assigned a corresponding slot. If the number of slots is greater than the number of objects, then some of the slots remain empty (do not contain objects). This approach showed significant results in such object-centric tasks as set property prediction and object discovery.

The iterative slot-based approach can be considered as a variant of the soft K-Means algorithm (Bauckhage (2015b)), where the key/value/query projections are replaced with the identity function and updating via the recurrent neural network is excluded (Locatello et al. (2020)). In our work, we propose another version of the generalization, when the K-Means algorithm is considered as a Gaussian Mixture Model (Bauckhage (2015a)). We represent slots not only as centers of clusters, but we also use information about the distance between clusters and assigned vectors, which leads to more expressive slot representations. Representing slots in this way improves the quality of the model in object-centric problems, achieving the state-of-the-art results in the set property prediction task, even compared to highly specialized models (Zhang et al. (2019b)), and also improves the generalization ability of the model for image reconstruction task.

The paper is structured as follows. In Section 2, we provide background information about Slot Attention module and Mixture Models. We also describe the process of their training. In Section 3, we introduce a Slot Mixture Module — the modification of a Slot Attention Module which provides

more expressive slot representations. In Section 4.1, by extensive experiments we show that the proposed Slot Mixture Module riches the state-of-the-art performance in the set property prediction task on the CLEVR dataset Johnson et al. (2017) and outperforms even highly specialized models Zhang et al. (2019b). In Section 4.2, we provide experimental results for the image reconstruction task on four datasets: three with synthetic images (CLEVR-Mirror Singh et al. (2021a), ShapeStacks Groth et al. (2018b), ClevrTex Karazija et al. (2021)) and one with real life images COCO-2017 (Lin et al. (2014)). The proposed Slot Mixture Module improves reconstruction performance. In Section 4.3, we demonstrate that Slot Mixture Module outperforms original Slot Attention on the Object Discovery task on the CLEVR10 dataset. In Section 4.4, we compare K-Means and Gaussian Mixture Model clustering approaches on the Set Property Prediction task and show that Gaussian Mixture Model clustering is a better choice for object-centric learning. We give a short overview of related works in Section 5. In Section 6, we discuss the obtained results, advantages, and limitations of our work.

The main contributions of our paper are as follows:

1. We proposed a generalization of slot-based approach for object-centric representations as a Gaussian Mixture Model (Section 3).

2. Such a representation allows state-of-the-art performance in the set property prediction task, even in comparison with specialized models (Section 4.1), which are not aimed at building disentangled representations.

3. The slot representations as a Gaussian Mixture improve the generalization ability of the model in other object-oriented tasks (Section 4.2).

4. The Slot Mixture Module shows a much faster convergence on the Object Discovery task compare to the Original Slot Attention (Section 4.3).

## 2 BACKGROUND

### 2.1 SLOT ATTENTION

Slot Attention (SA) module (Locatello et al. (2020)) is an iterative attention mechanism that is designed to map a distributed feature map to a set of $K$ slots. Randomly initialized slots from a Gaussian distribution with trainable parameters are used to get $q$ projections of slots. Feature map vectors with corresponding projections serve as $k$ and $v$ vectors. Dot-product attention between $q$ and $k$ vectors with the softmax across $q$ dimension implies competition between slots for explaining parts of the input. Attention coefficients are used to assign $v$ vectors to slots via a weighted mean.

$$M = \frac{1}{\sqrt{D}} k(\text{inputs}) q(\text{slots})^T \in \mathbb{R}^{N \times K}, \quad \text{attn}_{i,j} = \frac{e^{M_{i,j}}}{\sum_{j=1}^{K} e^{M_{i,j}}},$$

$$W_{i,j} = \frac{\text{attn}_{i,j}}{\sum_{i=1}^{N} \text{attn}_{i,j}}, \quad \text{updates} = W^T v(\text{inputs}) \in \mathbb{R}^{K \times D}$$

Gated Recurrent Unit (GRU) (Cho et al. (2014)) is used for addition slots update. It takes slot representations before update iteration as a hidden state and updated slots as inputs. The important property of Slot Attention is that it has permutation invariance with respect to input vectors of the feature map and permutation equivariance with respect to slots. These properties make the Slot Attention module suitable for operating with sets and object-centric representations.

Technically, Slot Attention is a learnable analogue of K-Means clustering algorithm with an additional trainable GRU update step and dot product (with trainable $q$, $k$, $v$) projections instead of Euclidean distance as the measure of similarity between the input vectors and cluster centroids. At the same time, K-Means clustering can be considered as a simplified version of the Gaussian Mixture Model.

## 2.2 Mixture Models

Mixture Models (MM) is a class of parametric probabilistic models, in which it is assumed that each $\boldsymbol{x}_i$ from some observations $\boldsymbol{X} = \{\boldsymbol{x}_1, ..., \boldsymbol{x}_N\} \in \mathbb{R}^{N \times D}$ is sampled from the mixture distribution with $K$ mixture components and prior mixture weights $\boldsymbol{\pi} \in \mathbb{R}^K$:

$$\boldsymbol{x}_i \sim p(\boldsymbol{x}_i|\boldsymbol{\theta}) = \sum_{k=1}^K \pi_k p(\boldsymbol{x}_i|\boldsymbol{\theta}_k), \ P(\boldsymbol{X}|\boldsymbol{\theta}) = \prod_{i=1}^N p(\boldsymbol{x}_i|\boldsymbol{\theta}), \ \sum_k \pi_k = 1.$$

These models can be seen as the models with latent variables $z_{i,k} \in \{z_1, ..., z_K\}$ that indicate which component $\boldsymbol{x}_i$ came from. The problem is to find such $K$ groups of component parameters $\boldsymbol{\theta}_k$ and component assignments of each sample $\boldsymbol{x}_i$ that will maximize the likelihood of the model $P(\boldsymbol{X}|\boldsymbol{\theta})$. The Expectation Maximization (EM) algorithm is an iterative algorithm that addresses this problem and includes two general steps. The **Expectation (E) step**: evaluate the expected value of the complete likelihood $P(\boldsymbol{X}, \boldsymbol{Z}|\boldsymbol{\theta}^*)$ with respect to the conditional distribution of $P(\boldsymbol{Z}|\boldsymbol{X}, \boldsymbol{\theta})$:

$$Q(\boldsymbol{\theta}^*, \boldsymbol{\pi}^*|\boldsymbol{\theta}, \boldsymbol{\pi}) = \mathbb{E}_{P(\boldsymbol{Z}|\boldsymbol{X}, \boldsymbol{\theta})}[\log P(\boldsymbol{X}, \boldsymbol{Z}|\boldsymbol{\theta}^*)], \ P(\boldsymbol{X}, \boldsymbol{Z}|\boldsymbol{\theta}^*) = \prod_{i=1}^N \prod_{k=1}^K [\pi_k p(\boldsymbol{x}_i|\boldsymbol{\theta}_k^*)]^{I(z_i=z_k)},$$

where $I(*)$ is an indicator function.

The **Maximization (M) step**: find $\boldsymbol{\theta}^*, \boldsymbol{\pi}^*$ that maximize $Q(\boldsymbol{\theta}^*, \boldsymbol{\pi}^*|\boldsymbol{\theta}, \boldsymbol{\pi})$:

$$(\boldsymbol{\theta}, \boldsymbol{\pi}) = \operatorname{argmax}_{(\boldsymbol{\theta}^*, \boldsymbol{\pi}^*)} Q(\boldsymbol{\theta}^*, \boldsymbol{\pi}^*|\boldsymbol{\theta}, \boldsymbol{\pi}).$$

One of the most widely used models of this kind is Gaussian Mixture Model (GMM), where each mixture component is modeled as a Gaussian distribution parameterized with its mean values and covariance matrix, which is diagonal in the simplest case: $P(\boldsymbol{x}_i|\boldsymbol{\theta}_k) = \mathcal{N}(\boldsymbol{x}_i|\boldsymbol{\mu}_k, \boldsymbol{\Sigma}_k)$, $\boldsymbol{\Sigma}_k = \operatorname{diag}(\boldsymbol{\sigma}_k^2)$. In this case, EM algorithm is reduced to the following calculations.

**E step**:

$$p(z_k|\boldsymbol{x}_i) = \frac{p(z_k)p(\boldsymbol{x}_i|\boldsymbol{\theta}_k)}{\sum_{k=1}^K p(z_k)p(\boldsymbol{x}_i|\boldsymbol{\theta}_k)} = \frac{\pi_k \mathcal{N}(\boldsymbol{x}_i|\boldsymbol{\mu}_k, \boldsymbol{\Sigma}_k)}{\sum_{k=1}^K \pi_k \mathcal{N}(\boldsymbol{x}_i|\boldsymbol{\mu}_k, \boldsymbol{\Sigma}_k)} = \gamma_{k,i}.$$

**M step**:

$$\pi_k^* = \frac{\sum_{i=1}^N \gamma_{k,i}}{N}, \ \boldsymbol{\mu}_k^* = \frac{\sum_{i=1}^N \gamma_{k,i}\boldsymbol{x}_i}{\sum_{i=1}^N \gamma_{k,i}}, \ \boldsymbol{\Sigma}_k^* = \frac{\sum_{i=1}^N \gamma_{k,i}(\boldsymbol{x}_i - \boldsymbol{\mu}_k)(\boldsymbol{x}_i - \boldsymbol{\mu}_k)^T}{\sum_{i=1}^N \gamma_{k,i}}.$$

The key difference between the Gaussian Mixture Model and K-Means clustering is that GMM considers not only the centers of clusters, but also the distance between clusters and assigned vectors with the prior probabilities of each cluster.

## 3 Slot Mixture Module

For the purposes of the object-centric learning we propose a modified Gaussian Mixture Model approach and call it the Slot Mixture Module (SMM). This module uses GMM **E** and **M steps** (Section 2.2) to map feature map vectors from the convolutional neural network (CNN) encoder to the set of slot representations, where slots are concatenation of mean values and diagonal of covariance matrix. This set of slots is further used in the downstream task. We also use the same additional neural network update step for the mean values before updating the covariance values:

$$\boldsymbol{\mu}_k = \operatorname{RNN}(\text{input=}\boldsymbol{\mu}_k^*, \ \text{hidden=}\boldsymbol{\mu}_k), \ \boldsymbol{\mu}_k = \operatorname{MLP}(\boldsymbol{\mu}_k) + \boldsymbol{\mu}_k.$$

These two steps serve the needs of the downstream task by linking the external and internal models. The internal model (E and M steps in SMM) tries to update its parameters $\boldsymbol{\mu}, \boldsymbol{\Sigma}$ so that the input vectors $x$ are assigned to slots with the maximum likelihood, while external model takes these parameters as input. The full pseudocode is presented in the Algorithm 1. A function $f_\theta(\boldsymbol{x}, \boldsymbol{\mu}, \boldsymbol{\Sigma}_{diag})$ : $\mathbb{R}^{N \times D} \times \mathbb{R}^{K \times D} \to \mathbb{R}^{N \times K}$ stands for a log-Gaussian density function with additional $k$ and $q$ projections for input and slot vectors.

Slot Mixture Module can be seen as an extension of the Slot Attention Module with the following key differences: (1) SMM updates not only the mean values, but also the covariance values and prior probabilities, (2) the Gaussian density function is used instead of the dot-product attention, and (3) slots are considered not only as mean values of the cluster, but as the concatenation of mean and covariance values.

---

**Algorithm 1:** The Slot Mixture Module pseudocode. $\boldsymbol{\pi}$ is initialized as uniform categorical distribution, $\boldsymbol{\mu}$ and $\boldsymbol{\Sigma}_{diag}$ are initialized from Gaussian distributions with trainable parameters.

---

**Input:** $\boldsymbol{x} \in \mathbb{R}^{N \times D}$ — flattened CNN feature map with added positional embeddings;
$\quad\quad \boldsymbol{\mu}, \boldsymbol{\Sigma}_{diag} \in \mathbb{R}^{K \times D}, \boldsymbol{\pi} \in \mathbb{R}^K$ — SMM initialization parameters.

**Output:** slots $\in \mathbb{R}^{N \times 2D}$.

1  $\boldsymbol{x} = \text{MLP}(\text{LayerNorm}(\boldsymbol{x}))$
2  **for** $t = 0...T$ **do**
3     logits = $f_\theta(\boldsymbol{x}, \boldsymbol{\mu}, \boldsymbol{\Sigma}_{diag})$
4     gammas = SoftMax(logits + $\log \boldsymbol{\pi}$, dim=1)
5     $\boldsymbol{\pi}$ = gammas.mean(dim=0)
6     $\boldsymbol{\mu}^* = \text{WeightedMean}(\text{weights=gammas, values=}\boldsymbol{x})$
7     $\boldsymbol{\mu} = \text{GRU}(\text{input=}\boldsymbol{\mu}^*, \text{hidden=}\boldsymbol{\mu})$
8     $\boldsymbol{\mu} \mathrel{+}= \text{MLP}(\text{LayerNorm}(\boldsymbol{\mu}))$
9     $\boldsymbol{\Sigma}_{diag} = \text{WeightedMean}(\text{weights=gammas, values=}(\boldsymbol{x} - \boldsymbol{\mu})^2))$
10 slots = concat($[\boldsymbol{\mu}, \boldsymbol{\Sigma}_{diag}]$)
11 **return** slots

---

# 4 EXPERIMENTS

Since our model can be seen as an extension of the Slot Attention, we use it as our main competitor. In each experiment, we trained two versions of the same model: the one with the Slot Attention module and another one with our SMM, keeping the same training conditions. We also make use of the technique from (Chang et al. (2022)) detaching slots from the gradient computational graph at the last iteration of the algorithm in every experiment.

With equal dimensionality of input vectors, SMM implies twice the dimensionality of the slots in comparison with SA (SA represents slots with $\boldsymbol{\mu}$, while SMM uses concatenation of $\boldsymbol{\mu}, \boldsymbol{\Sigma}_{diag}$). To ensure a fair comparison of approaches, we added matrix multiplication after the SMM module, reducing the dimensionality by a factor of two.

## 4.1 SET PROPERTY PREDICTION

Neural networks for sets are involved in various applications across many data modalities (Carion et al. (2020); Achlioptas et al. (2017); Simonovsky & Komodakis (2018); Fujita et al. (2019)). Set Property Prediction is a supervised task that requires the model to predict an unordered set of vectors representing the properties of objects from the input image. Sets of predicted and target vectors are matched using a Hungarian algorithm (Kuhn (1955)) and the learning signal is provided by Huber Loss (Zhang et al. (2019a)) between matched vectors. Slot Mixture Module is suitable for operating with sets as it preserves permutation equivariance regarding mixture components (slots) and initializes them randomly. The scheme of the model is shown in the Fig. 1.

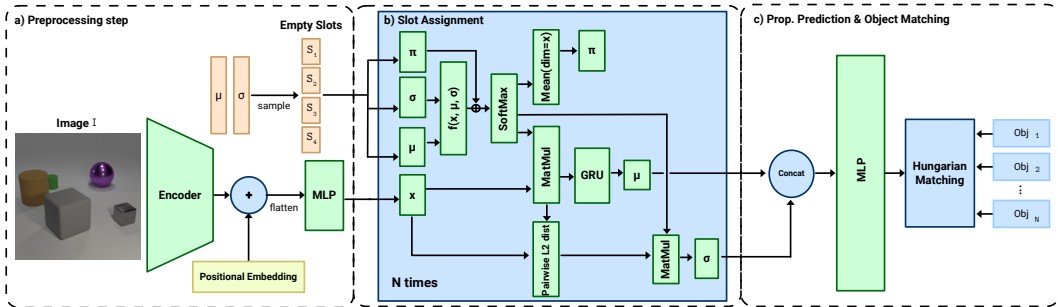

Figure 1: Architecture of the set property prediction model. Input image is encoded into a set of vectors with added positional embeddings via CNN encoder. These vectors are assigned to slots with our Slot Mixture Module. Then slots representations are passed to the MLP to predict object properties. Ground truth and predicted sets are matched using the Hungarian algorithm.

**Setup**  We use the CLEVR dataset with rescaled images to a resolution of $128 \times 128$ as the data source. All modules except SA/SMM are the same as in the (Locatello et al. (2020)). Each model is trained with Adam optimizer (Kingma & Ba (2015)) for $1.5 \times 10^5$ iterations with OneCycleLR (Smith & Topin (2019)) learning rate scheduler at a maximum learning rate of $4 \times 10^{-4}$, we use a batch size of $512$. SA/SMM number of iterations is set to $5$ during training and to $7$ during evaluation. The number of slots is equal to $10$ since CLEVR images contain $10$ or fewer objects.

**Results**  Quantitative results of the experiments are presented in Table 1. We compute Average Precision (AP) with a certain distance threshold ($\infty$ means we do not use distance threshold). A predicted vector of properties and coordinates is correct if there is an object with the same properties within the threshold distance. The lower the distance threshold, the more difficult the task. Our experiments demonstrate it is possible to improve significantly the original Slot Attention performance via detaching slots before the last iteration of slots refinement and rescaling coordinates to a wider range of values. With these modifications, the original Slot Attention performance is still worse than the current state-of-the-art model iDSPN, while our SMM confidently outperforms iDSPN.

Table 1: Set prediction performance on the CLEVR dataset (AP in %, mean ± std for 4 seeds in our experiments). Slot Attention (Locatello et al. (2020)) and iDSPN (Zhang et al. (2019b)) results are from the original papers. SA* is the Slot Attention model that was trained with the same conditions as our SMM: detached slots at the last iteration, rescaled coordinates to the range of [-1, 1].

| Model | $AP_\infty$ | $AP_1$ | $AP_{0.5}$ | $AP_{0.25}$ | $AP_{0.125}$ | $AP_{0.0625}$ |
|---|---|---|---|---|---|---|
| SA | $94.3 \pm 1.1$ | $86.7 \pm 1.4$ | $56.0 \pm 3.6$ | $10.8 \pm 1.7$ | $0.9 \pm 0.2$ | - |
| SA* | $97.1 \pm 0.7$ | $94.5 \pm 0.7$ | $88.3 \pm 3.2$ | $62.5 \pm 5.4$ | $23.6 \pm 1.4$ | $4.6 \pm 0.3$ |
| iDSPN | $98.8 \pm 0.5$ | $98.5 \pm 0.6$ | $98.2 \pm 0.6$ | $95.8 \pm 0.7$ | $76.9 \pm 2.5$ | $32.3 \pm 3.9$ |
| SMM (ours) | $\mathbf{99.4 \pm 0.2}$ | $\mathbf{99.3 \pm 0.2}$ | $\mathbf{98.8 \pm 0.4}$ | $\mathbf{98.4 \pm 0.7}$ | $\mathbf{92.1 \pm 1.2}$ | $\mathbf{47.3 \pm 2.5}$ |

## 4.2 Image Reconstruction Using Transformer

For comparison in an unsupervised image-to-image task, we use the SLATE model (Singh et al. (2021a)) replacing the Slot Attention module with our SMM. Unlike the pixel-mixture decoders, SLATE uses an Image GPT (Chen et al. (2020)) decoder conditioned on slot representations to reconstruct in the autoregressive manner the discrete visual tokens from a discrete VAE (dVAE) (Im et al. (2017)) treating pre-computed slot representations from a dVAE output as query vectors and latent code-vectors of the image as key/value vectors. SLATE demonstrates an impressive ability to capture complex interactions between the slots and pixels of the synthetic images, but exhibits poor performance for real-world data. dVAE encoder, decoder, and latent discrete tokens are receiving training signals from MSE between the input image and the reconstructed one with dVAE only. SA/SMM modules and Image GPT are trained with a cross-entropy using compressed image into

dVAE tokens as a target distribution, these gradients are blocked from the rest of the model (i.e. dVAE), but both parts of the system are trained simultaneously.

**Setup**    We consider the following datasets: CLEVR-Mirror (Singh et al. (2021a)), ClevrTex (Karazija et al. (2021)), ShapeStacks (Groth et al. (2018b)), and COCO-2017 (Lin et al. (2014)). CLEVR-Mirror is an extension of the standard CLEVR dataset, which requires capturing global relations between local components due to the presence of a mirror, ShapeStacks tests the ability of the model to describe complex local interactions (multiple different objects stacked on each other), ClevrTex examines the capabilities of the model in the textural-rich scenes. For ShapeStacks, ClevrTex and COCO we used images rescaled to the resolution of $96 \times 96$, CLEVR-Mirror images are rescaled to $64 \times 64$. Training conditions with hyperparameters corresponding to a certain dataset are taken from (Singh et al. (2021a)), except that we use a batch size equal to $64$ and $2.5 \times 10^5$ training iterations for all experiments.

**Results**    Table 2 shows the metrics of the image reconstruction performance for the test-parts of different datasets. We use mean squared error (MSE) and Frechet Inception Distance (FID) (Heusel et al. (2017)) computed with the PyTorch-Ignite library (Fomin et al. (2020)) as measure of quality of the generated images. We also evaluate cross-entropy between tokens from discrete VAE (as one-hot distributions) and predicted distributions by Image GPT that are conditioned by certain slot representations, as MSE and FID can be limited by the discrete VAE.

Table 2: Reconstruction performance using Image GPT decoder with different conditions.

| Data | FID | | MSE | | Cross-Entropy | |
|---|---|---|---|---|---|---|
| | SA | SMM | SA | SMM | SA | SMM |
| CLEVR-Mirror | 35.4 | 34.8 | 4.5 | 4.3 | 0.82 | **0.20** |
| ShapeStacks | 56.6 | **50.4** | 102.2 | **67.3** | 88.3 | **66** |
| ClevrTex | 116 | 113 | 358 | 344 | 566 | **517** |
| COCO | 129 | **122** | 1354 | **938** | 540 | **479** |

Fig. 2 indicates the advantage of SMM in terms of cross-entropy during training across all the datasets we use. But as can be seen from the Table 2, this advantage does not always translate into much better image quality metrics of generated images due to dVAE limitations. The largest increase in quality was obtained for the ShapeStacks dataset. Examples of the images from ShapeStacks, ClevrTex and CLEVR-Mirror datasets and their reconstructions are presented in the Fig. 3. The model trained with SMM is much more likely to reconstruct the correct order of objects from the original image, as can be seen in examples from the random batch of 64 images from ShapeStacks dataset (see Fig. 4).

Even though we were able to improve the quality of real-world object-centric image generation, the final quality in general is quite poor and attention maps barely reflect human object-centered vision (see Fig. 5). Scaling and extending such visual models to handle a wide range of complex real-world images is an area of the future research.

## 4.3    OBJECT DISCOVERY

Another unsupervised object-centric image-to-image task is the Object Discovery. In this task, each slot representation is decoded into the 4-channel image using a Spatial Broadcast decoder (Watters et al. (2019b)). The resulting reconstruction is estimated as a mixture of decoded slots, where the

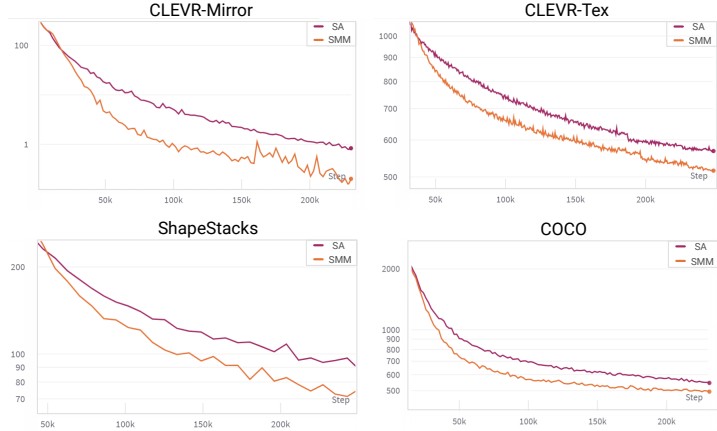

Figure 2: Validation cross-entropy during training for 4 different datasets. Our experiments show that using SMM module instead of SA improves validation performance of the autoregressive transformer by about 10 percent during training. The result is maintained for all the datasets that we use.

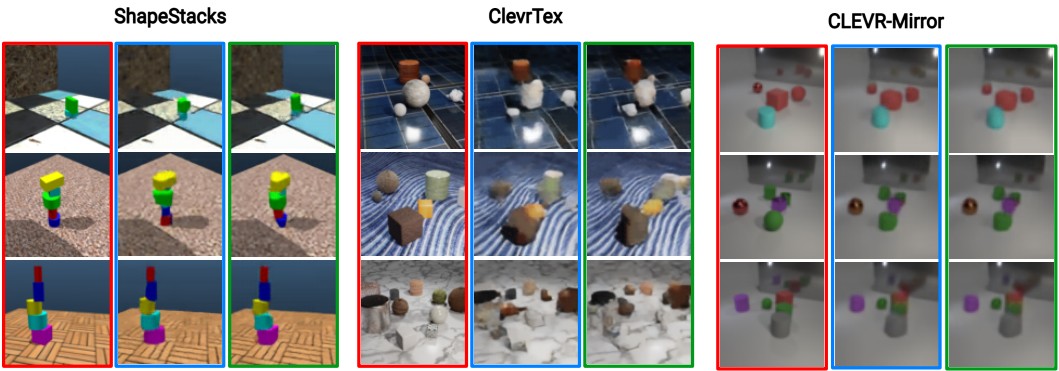

Figure 3: Examples of image generation with Image GPT conditioned to different slot representations. Images in the blue borders are from the model with the Slot Attention module, and images in green borders are generated using slots from the Slot Mixture Module. Red color stands for input images.

first three channels are responsible for the reconstructed RGB image and the fourth channel is for the weights of the mixture component.

**Setup** To compare with the Slot Attention module, we consider the same training setup from the original work (Locatello et al. (2020)) for the CLEVR10 dataset but with the decreased number of training steps (300k instead of 500k).

**Results** Table 3 shows similarity between ground truth segmentation masks of objects (excluded background) and mixture coefficients estimated via the Adjusted Rand Index (ARI) score. Fig.6 demonstrates a much faster convergence of the model that uses SMM instead of the Slot Attention, which results in higher ARI score.

## 4.4 COMPARING VANILLA CLUSTERING

Slot Attention and Slot Mixture modules can be reduced to K-Means and Gaussian Mixture Model clustering approaches by removing GRU/MLP updates, trainable $q, k, v$ projections and LayerNorm layers (Ba et al. (2016)). Table 4 shows results of training set prediction model for CLEVR dataset

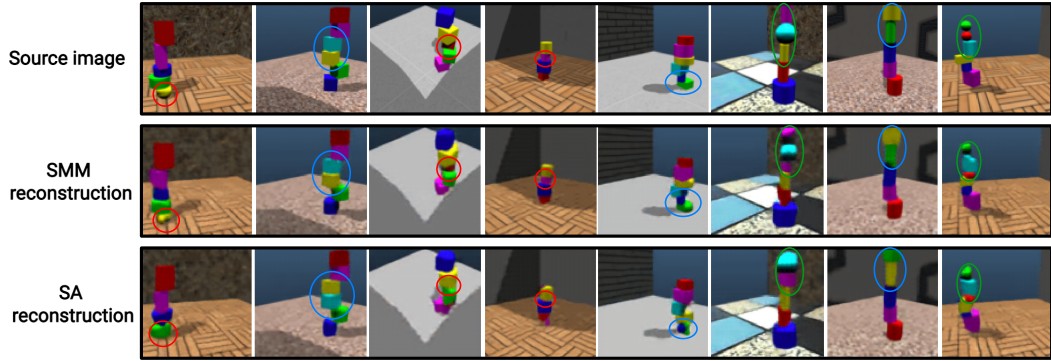

Figure 4: Examples of all the qualitatively incorrectly generated images from the random batch of 64 samples. In 6 cases reconstruction using Slot Attention gave a wrong order of objects (blue circle) or lost 1 object (red circle), in the remaining 2 samples both modules gave incorrect reconstruction (green circle).

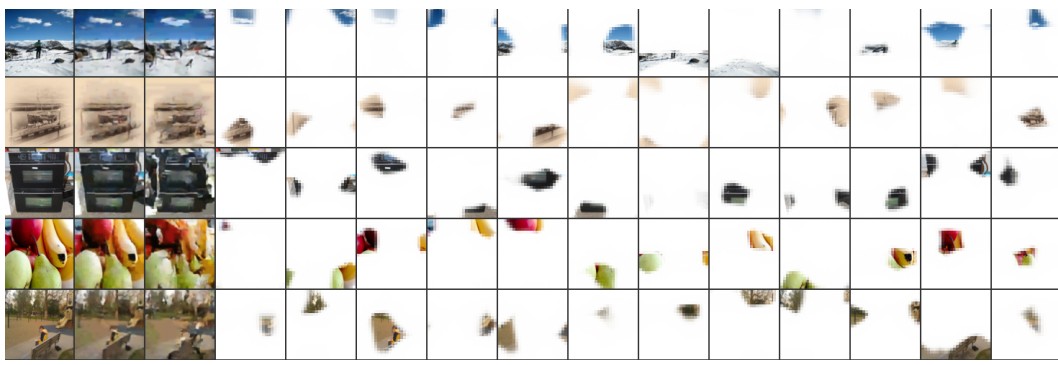

Figure 5: Examples of COCO image generations with Image GPT conditioned to slot representations from SMM. The first and the second columns show the input images and the end-to-end dVAE reconstruction correspondingly. The third column shows the generated image and all other columns demonstrate the corresponding attention maps from slots to the input image.

using these vanilla clustering methods. Our experiments demonstrate that GMM clustering is a better choice for object-centric learning, even without trainable layers.

Table 3: Adjusted Rand Index (ARI) score between ground truth masks of objects and mixture coefficients for CLEVR10 after 300k iterations of training.

| Model | ARI |
| --- | --- |
| SA | 85.9 |
| SMM | **91.3** |

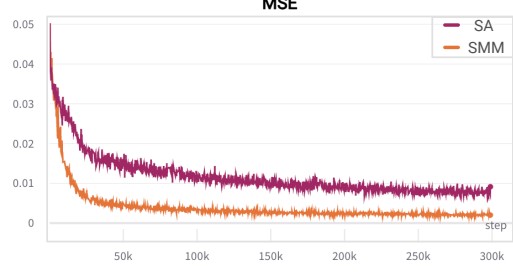

Figure 6: Mean Squared Error (MSE) between input and reconstructed images for Object Discovery task during training.

## 5 RELATED WORKS

**Set prediction** Neural network models for sets are applied to different machine learning tasks, such as point cloud generation (Achlioptas et al. (2017)), object detection (Carion et al. (2020)), speaker diarization (Fujita et al. (2019)), and molecule generation (Simonovsky & Komodakis (2018)). Albeit the set structure is suitable in many cases, traditional deep learning models are not inherently suitable for representing sets. There are some approaches that are built to reflect the unordered nature of

Table 4: Average Precision with different distance thresholds for set property prediction task on the CLEVR dataset after 100k iterations of training.

| Model | $AP_\infty$ | $AP_1$ | $AP_{0.5}$ | $AP_{0.25}$ | $AP_{0.125}$ |
|---|---|---|---|---|---|
| K-Means | 81.7 | 49.1 | 7.2 | 1.4 | 0.2 |
| GMM | **88.6** | **53.3** | **9.2** | **2.3** | **0.5** |

sets, e.g., the Deep Set Prediction Network (DSPN) (Zhang et al. (2019a)) reflects permutation symmetry by running an inner gradient descent loop that changes a set to encode more similarly to the input. An improved version of DSPN — iDSPN (Zhang et al. (2019b)) with approximate implicit differentiation provides better optimizations with faster convergence and state-of-the-art performance on the CLEVR dataset. Such models as Slot Attention and TSPN (Kosiorek et al. (2020)) use set-equivariant self-attention layers to represent the structure of sets.

**Object-centric representation** The discovery of objects in a scene in an unsupervised manner is a crucial aspect of representation learning, especially for object-centric tasks. In recent years, many approaches have been presented to solve this problem (Greff et al. (2019); Burgess et al. (2019); Engelcke et al. (2020; 2021) and others). Such methods as IODINE (Greff et al. (2019)), MONET (Burgess et al. (2019)), and GENESIS (Engelcke et al. (2020)) are built upon the Variational Autoencoder (VAE) framework (Kingma & Welling (2014); Rezende et al. (2014)). MONET uses the attention network that generates masks and conditions VAE on these masks. IODINE models an image as a spatial Gaussian mixture model to jointly infer the object representation and segmentation. Compared to MONET and IODINE, GENESIS explicitly models dependencies between scene components that allow the sampling of novel scenes. MONET, IODINE, and GENESIS use multiple steps to encode and decode an image, while Slot Attention (and its sequential extension for video (Kipf et al. (2021))) uses one step but performs an iterative procedure inside this step. The useful property of Slot Attention is that it produces the set of output vectors (slots) with permutation symmetry. Slots group input information and could be used in unsupervised tasks (object discovery) and supervised tasks (set prediction). GENESIS-v2 (Engelcke et al. (2021)), a development of the GENESIS model, uses attention masks similarly to (Locatello et al. (2020)).

## 6 CONCLUSION AND DISCUSSION

In this paper, we propose a new slot-based approach for object-centric representations — the Slot Mixture Module. Our module is a generalization of the K-Means clustering algorithm to the Gaussian Mixture Model (Bauckhage (2015a)). Unlike other approaches, we represent slots not only as centers of clusters, but we also use information about the distance between clusters and assigned vectors, which leads to more expressive slot representations. We have demonstrated on the CLEVR dataset (Johnson et al. (2017)) that using this module achieves the best results in the set property prediction task, even compared to highly specialized models. Also, the use of Slot Mixture Module shows considerable results in the image reconstruction task. On synthetic datasets, such as CLEVR-Mirror (Singh et al. (2021a)), ShapeStacks (Groth et al. (2018b)), ClevrTex (Karazija et al. (2021)), we achieved improved reconstruction performance. However, it is worth noting that modern object-centric models still do not perform well enough on real-life images, such as in COCO-17 (Lin et al. (2014)). The generalization of such methods to real data is an important scientific problem. We also show that Slot Mixture Module outperforms original Slot Attention on the Object Discovery task on the CLEVR10 dataset.

Also, in models using the slot representation, the number of slots is a hyperparameter, the same as the number of clusters for the K-Means clustering algorithm. In object-centric methods, this parameter is usually chosen equal to the maximum number of objects in the image plus one (assuming one slot is reserved for the background). This value varies from image to image, making it difficult to transfer from one dataset to another. It should be noted that, in general, the issue of transfer learning for slot-based models has not been sufficiently studied. Another limitation of modern slot-based models is that they work with a small number of slots (about 10), which means a small number of objects in the image. Therefore, the issue of scalability of such models to a larger number of objects is considered as a direction for further research.

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

# A   ARCHITECTURE DETAILS

Tables 5, 6, 7 describe hyperparameters for our experiments. In the case of using SMM instead of SA we use an additional dimensionality reduction for slots via trainable matrix multiplication.

Table 5: Architecture of the CNN encoder for the experiments on CLEVR dataset for set property prediction and object discovery tasks. Set prediction uses stride of 2 in the layers with *, while object discovery model uses stride of 1 in these layers.

| Layer | Channels | Activation | Params |
|---|---|---|---|
| Conv2D 5 ×5 | 64 | ReLU | stride: 1 |
| Conv2D 5 ×5 | 64 | ReLU | stride: 1/2* |
| Conv2D 5 ×5 | 64 | ReLU | stride: 1/2* |
| Conv2D 5 ×5 | 64 | ReLU | stride: 1 |
| Position Embedding | - | - | absolute |
| Flatten | - | - | dims: w, h |
| LayerNorm | - | - | - |
| Linear | 64 | ReLU | - |
| Linear | 64 | - | - |

Table 6: Spatial broadcast decoder for object discovery task.

| Layer | Channels/Size | Activation | Params |
|---|---|---|---|
| Spatial Broadcast | 8 ×8 | - | - |
| Position Embedding | - | - | absolute |
| ConvTranspose2D 5 ×5 | 64 | ReLU | stride: 2 |
| ConvTranspose2D 5 ×5 | 64 | ReLU | stride: 2 |
| ConvTranspose2D 5 ×5 | 64 | ReLU | stride: 2 |
| ConvTranspose2D 5 ×5 | 64 | ReLU | stride: 2 |
| ConvTranspose2D 5 ×5 | 64 | ReLU | stride: 1 |
| ConvTranspose2D 3 ×3 | 4 | - | stride: 1 |
| Split Channels | RGB (3), mask (1) | Softmax on masks (slots dim) | - |
| Combine components | - | - | - |

Table 7: Hyperparameters used for our experiments with SLATE architecture.

| Module | Parameter | Value |
|---|---|---|
|  | Image Size | 96 |
|  | Encoded Tokens | 576 |
| dVAE | Vocab size | 4096 |
| dVAE | Temp. Cooldown | 1.0 to 0.1 |
| dVAE | Temp. Cooldown Steps | 30000 |
| dVAE | LR (no warmup) | 0.0003 |
| Transformer | Layers | 8 |
| Transformer | Heads | 8 |
| Transformer | Hidden Dim. | 192 |
| SA/SMM | Num. slots | 12 |
| SA/SMM | Iterations | 7 |
| SA/SMM | Slot dim. | 192 |

