# OpenReview forum: "Object-Centric Learning with Slot Mixture Models"
_ICLR.cc/2023/Conference — Submitted to ICLR 2023_

### Official Review · Reviewer_LBbF · 2022-10-23

**Confidence:** 4
**Correctness:** 3
**Technical Novelty And Significance:** 2
**Empirical Novelty And Significance:** 3
**Recommendation:** 5

**Clarity, Quality, Novelty And Reproducibility:**

The overall presentation is quite terse, with little explanation regarding the motivation for the method and its design choices. The argument that SMM's improvements over slot attention stem from it taking into account distances between slots and input features could use some elaboration - which distance metric is meant by this, and why can't slot attention simulate similar behavior?

The text leans heavily on the original slot attention paper, and would be difficult to follow without having read it first. Adding a small section introducing the problem statement, and slot attention, would make the paper more self contained. In general, the writing would benefit from another editing pass, as it contains a number of errors.

Some implementation details are missing, most importantly the definition of $f_\theta(x, \mu, \Sigma)$, but also the architecture of the MLP updating $\mu$. This impairs the paper's reproducability. While these definitions should be included in any case, this issue could also be alleviated if the authors choose to publish their submitted code.




**Strength And Weaknesses:**

Strengths:
 - The paper addresses the interesting and relevant problem of improving understanding and performance of the slot attention algorithm.
 - The connection to GMMs is interesting and generally sensible.
 - The set prediction results on CLEVR are promising, especially at lower distance thresholds.

Weaknesses:
 - The paper's contribution is quite incremental, especially since the connection to soft k-means was already discussed in the slot attention paper.
 - The centerpiece of the GMM, the likelihood model, is abstracted away into the learned function $f_\theta(x, \mu, \Sigma)$, which is never defined in detail. This not only hurts reproducability, but also makes it difficult to understand to what degree the resulting system still resembles a GMM.
 - In general, the motivating connection to GMMs is underexplored. The proposed method appears to be a somewhat ad-hoc combination of the GMM formulas and learned attention/GRU components as in slot attention. It is unclear which properties of either system transfer to SMM. It would at least be useful to include experimental results for a "pure GMM" version of the model, similar to what was done in the Appendix of the slot attention paper for k-means.
 - The image reconstruction experiment is not convincing. The motivation behind object-centric models like slot attention or SLATE is to learn compositional representations by introducing representation bottlenecks, typically at the cost of reducing reconstruction quality. Exclusively measuring reconstruction quality (and related metrics), without also evaluating the usefulness of the learned representation, therefore misses the point - such improvements could be easily achieved by removing bottlenecks. Given the unconvincing segmentations visible in Fig. 3, it seems plausible that the improvements in reconstruction quality came at the expense of the models primary purpose, the learning of compositional representations.
- It is unclear why the two-stage model SLATE is chosen as a baseline, as the dVAE preprocessing seems to introduce addtional complications making the results harding to interpret.
- The clarity of the writing could be significantly improved (see below).


**Summary Of The Paper:**

The paper proposes a variant of slot attention resembling the expectation-maximization (EM) algorithm for learning Gaussian mixture models: Slots are represented by a set of means and isotropic variances, and they are inferred using EM update rules combined with some learned mappings. It is shown that the resulting component, named slot mixture model (SMM), yields improved set prediction results on CLEVR, as well as improved reconstruction quality on multi object image datasets preprocessed via a discrete variational autoencoder (dVAE).

**Summary Of The Review:**

Overall, the proposed a slot attention variant shows promising set prediction results. However, the presentation, analysis, and representation learning evaluation could all be significantly improved. As a result, I do not think the paper is ready for publication in its current form.

----

Post rebuttal update: The authors have introduced major changes to the paper, which partially address my concerns. I think the vanilla clustering and object discovery experiments are both significant improvements, although the latter is somewhat basic and only conducted on one dataset. As expressed in my response to the authors, I still have some concerns regarding clarity and the setup of the SLATE experiment. Given this, as well as the magnitude of the changes introduced to the manuscript during the review process, I still think the paper is not quite ready for publication. I have increased my score from 3 to 5.

---

> ### Author Response · Authors · 2022-11-17
> **Response to Reviewer LBbF**
>
> Thank you for your comment. It is very helpful for us to continue to improve this work.
>
> **W1:** The paper's contribution is quite incremental, especially since the connection to soft k-means was already discussed in the slot attention paper.
>
> - **Re:** Object-centered learning based on slot attention is the focus of research. By now, several improvements to this mechanism have already been proposed, which we have listed in related works. It should be noted that many of them are incremental improvements from the point of view of algorithmic implementation, comprising one line of code. However, from the point of view of moving towards a more versatile approach and especially considering metrics on benchmarks, it is admitted that they all play an important role in the object-centric approach. Our idea is implemented quite simply, but seriously improves learning stability. Our approach also suggests looking at the slot competition process from a new point of view and proving that using SMM here gives results. SMM represents slots not only as centers of clusters, but also uses information about the distance between clusters and assigned vectors, which leads to more expressive slot representations.
>
> **W2:** The centerpiece of the GMM, the likelihood model, is abstracted away into the learned function , which is never defined in detail. This not only hurts reproducability, but also makes it difficult to understand to what degree the resulting system still resembles a GMM.
>
> - **Re:** Density function is just a log-Gaussian density function with additional k, q projections for slots and input vectors. Wa have added clear mentioning of this in the text (Section 3).
>
> **W3:** In general, the motivating connection to GMMs is underexplored. The proposed method appears to be a somewhat ad-hoc combination of the GMM formulas and learned attention/GRU components as in slot attention. It is unclear which properties of either system transfer to SMM. It would at least be useful to include experimental results for a "pure GMM" version of the model, similar to what was done in the Appendix of the slot attention paper for k-means.
>
> - **Re:** We have added a comparison between vanilla K-Means and GMM clustering approaches in the set property prediction task  (Table 4).
> |Model|$AP_\inf$|$AP_1$|$AP_{0.5}$|$AP_{0.25}$|$AP_{0.125}$|
> |-------|-----------|---------|-------------|--------------|---------------|
> |K-Means|81.7|49.1|7.2|1.4|0.2|
> |GMM |**88.6**|**53.3**|**9.2**|**2.3**|**0.5**|
>
> **W4:** The image reconstruction experiment is not convincing. The motivation behind object-centric models like slot attention or SLATE is to learn compositional representations by introducing representation bottlenecks, typically at the cost of reducing reconstruction quality. Exclusively measuring reconstruction quality (and related metrics), without also evaluating the usefulness of the learned representation, therefore misses the point - such improvements could be easily achieved by removing bottlenecks. Given the unconvincing segmentations visible in Fig. 3, it seems plausible that the improvements in reconstruction quality came at the expense of the models primary purpose, the learning of compositional representations.
>
> **W5:** It is unclear why the two-stage model SLATE is chosen as a baseline, as the dVAE preprocessing seems to introduce addtional complications making the results harding to interpret.
>
> - **Re:** We combined the answers to W4 and W5 into one. We chose the SLATE model to compare approaches, in particular, because it allows us to monitor the effectiveness of trained object representations not only through the reconstruction quality, but also through the proximity of the distributions predicted by the transformer for the dVAE token to the tokens from the dVAE encoder. In fact, the object-centric component here does not explicitly affect the dVAE encoder and decoder itself. The object-centricity here lies precisely in the transformer, which tries to predict a sequence that is a tokenized image by slots. These parts are trained simultaneously but separately from each other. The quantitative results of our experiments demonstrate that even in cases where there is no clear advantage in terms of reconstruction quality metrics, the transformer still more accurately predicts dVAE tokens, which indicates better and more expressive object representations in themselves (bottlenecks are still the same).
>
> **W6:** The clarity of the writing could be significantly improved (see below).
>
> - **Re:** We have done meticulous work to improve the quality and readability of the text. Spelling, grammatical and other errors were eliminated. We have added a detailed description of the function f in Section 3. We have also added a section 2.1 where a detailed description of Slot Attention is given.  In the Appendix A, we have given a detailed description of the architecture and details that were omitted in the main text of the paper.

---

> > ### Comment · Reviewer_LBbF · 2022-11-25
> > **Response**
> >
> > Thank you for your response and the updates to the paper.
> >
> > > Density function is just a log-Gaussian density function with additional k, q projections for slots and input vectors. Wa have added clear mentioning of this in the text (Section 3).
> >
> > This description still seems ambiguous to me. Slots consist of means and standard deviations; are both projected? If so, how are negative standard deviations prevented? In my opinion, for operations which are this central to the model, there is not good reason to not specify them exactly via equations. Also, the symbols k and q are never defined, and only make sense to the reader if they are familiar with the attention literature.
> >
> > > The quantitative results of our experiments demonstrate that even in cases where there is no clear advantage in terms of reconstruction quality metrics, the transformer still more accurately predicts dVAE tokens, which indicates better and more expressive object representations in themselves (bottlenecks are still the same).
> >
> > I am still not convinced by this. The DVAE is essentially a preprocessing step here (although jointly trained), so you are still measuring a sort of reconstruction quality, just in a hopefully more semantically meaningful latent space. While you are correct that the bottleneck remains the same in the sense that it has the same dimensionality as before, that doesn't mean that replacing the module inferring it doesn't hurt the system's compositional modelling ability. To put it simply, the goal of object-centric scene understanding is not to obtain the best possible reconstruction accuracy given a certain bottleneck dimensionality, but to infer useful compositional representations. In my opinion, evaluation should therefore always include metrics to that effect, e.g. ARI scores or a compositional generation experiment as in SLATE.

---

### Official Review · Reviewer_oeon · 2022-10-24

**Confidence:** 4
**Correctness:** 2
**Technical Novelty And Significance:** 3
**Empirical Novelty And Significance:** 2
**Recommendation:** 3

**Clarity, Quality, Novelty And Reproducibility:**

* Clarity: the paper could be more clear, as there are many typos and there are missing details. Citations do not follow the proper format (they are missing parenthesis in most cases).

* Quality: while the motivation behind the paper is clear, the presented algorithm is unclear and the evaluation is inconclusive.

* Originality: the motivation for the paper is novel enough.

* Reproducibility: Given the doubts about the quantitative results in the experiment section and the missing details on the method, I believe this paper would be hard to reproduce.

Example Typo - in the introduction section "in such object-centir" -> "in such object-centric"

**Strength And Weaknesses:**

**Strenghts**:

[+] Good quantitative results compared to Slot Attention in the reported benchmarks.

**Weaknesses**:

[-] Soundness: while the idea of using Gaussian mixture models for clustering features is interesting, in practice the authors perform a very different operation according to algorithm 1. Even though it is claimed that the model performs expectation-maximization, from the algorithm it does not seem like it. Additionally, the slots are represented as gaussians with a diagonal covariance, but it is not clear if the authors use the mean to represent the slots (making the variance redundant) or if they sample and then how would this sampling step impacts the gradient computation. In fact, the authors do not explictily mention what is the optimization objective and algorithm in the main document.

[-] Quality and clarity: the document is missing details that make it hard to assess its soundness as discussed in the previous point. Furthermore, the clarity of the document could be improved, with many typos and missing information.

[-] Qualitative results: the qualitative results hardly show a difference between the presented model and Slot Attention (making me doubt the quantitative claims). For example, for Figure 1 it is hard to see a qualitative difference. Furthermore, the results in Figure 3 are not informative, as they clearly show random segmentations of the input images (instead of object-centric segmentations) with poor quality reconstructions.

**Summary Of The Paper:**

The paper proposes an algorithm to extract object-centric representations from feature maps inspired by Slot Attention. The main insight is to   represent slots using a mixture of Gaussians. This comes from the observation that Slot Attention can be viewed as performing soft k-means clustering on features maps, and mixtures of Gaussians being another formulation to a similar problem. The authors show that their method is competitive with Slot Attention and other methods to extract object-centric representations.

**Summary Of The Review:**

While the paper has a clear and interesting motivation, the algorithm presented in the paper is unclear and has technical flaws. At the same time, the results are unconvincing, and therefore I argue for the rejection of the paper in its current state.

---

> ### Author Response · Authors · 2022-11-17
> **Response to Reviewer oeon**
>
> Thank you for the detailed review and insightful comments.
>
> **W1:** Soundness: while the idea of using Gaussian mixture models for clustering features is interesting, in practice the authors perform a very different operation according to algorithm 1. Even though it is claimed that the model performs expectation-maximization, from the algorithm it does not seem like it. Additionally, the slots are represented as gaussians with a diagonal covariance, but it is not clear if the authors use the mean to represent the slots (making the variance redundant) or if they sample and then how would this sampling step impacts the gradient computation. In fact, the authors do not explictily mention what is the optimization objective and algorithm in the main document.
>
> - **Re:** In Algorithm 1, lines 10 and 11 show that the slot representations are not just the average of the clusters, but the concatenation of the average and the diagonal of the covariance matrix. This is one of the key differences between GMM and Slot Attention. We do not use sampling in our approach.
> Algorithm 1 demonstrates the structure of a module that does not learn on its own, but is used as part of a model that uses object-oriented representations to solve a specific problem. In our work, we use the GMM module in two models: one solves the set property prediction problem, for which we use Huber Loss. The second is the SLATE model with the Slot Attention module replaced by GMM. We have also added an overview of the model architecture in Fig. 1.
>
> **W2:**  Quality and clarity: the document is missing details that make it hard to assess its soundness as discussed in the previous point. Furthermore, the clarity of the document could be improved, with many typos and missing information.
>
> -**Re:** We have done a thorough work to improve the quality and clarity of the text, grammatical and spelling errors have been eliminated, citations have been put in the proper form, incomprehensible places have been rewritten.
>
> **W3:**  Qualitative results: the qualitative results hardly show a difference between the presented model and Slot Attention (making me doubt the quantitative claims). For example, for Figure 1 it is hard to see a qualitative difference. Furthermore, the results in Figure 3 are not informative, as they clearly show random segmentations of the input images (instead of object-centric segmentations) with poor quality reconstructions.
>
> - **Re:** We have added Fig. 4 containing examples with more clear qualitative difference in ShapeStacks image generation
>
> **Comments on clarity, quality and reproducibility.**
>
> **Clarity:** the paper could be more clear, as there are many typos and there are missing details. Citations do not follow the proper format (they are missing parenthesis in most cases).
> - **Re:** As mentioned above, we carefully eliminated all spelling, grammatical and other errors in the text of the paper.
>
> **Quality:** while the motivation behind the paper is clear, the presented algorithm is unclear and the evaluation is inconclusive.
> - **Re:** Algorithm 1 demonstrates the structure of a module that does not learn on its own, but is used as part of a model that uses object-oriented representations to solve a specific problem. In our work, we use the GMM module in two models: one solves the set property prediction problem, for which we use Huber Loss. The second is the SLATE model with the Slot Attention module replaced by GMM. We have also added an overview of the model architecture in Figure 1. We have added Figure 4, which shows the differences in image restoration using our approach and SA.
>
> **Reproducibility:** Given the doubts about the quantitative results in the experiment section and the missing details on the method, I believe this paper would be hard to reproduce.
> - **Re:** We have added a detailed description of the model architecture and any missing details that prevent the reproducibility of the results in the Appendix A.

---

### Official Review · Reviewer_koSa · 2022-10-24

**Confidence:** 3
**Correctness:** 3
**Technical Novelty And Significance:** 3
**Empirical Novelty And Significance:** 2
**Recommendation:** 5

**Clarity, Quality, Novelty And Reproducibility:**

The clarity is lacking in certain regards, however, overall the paper was succinct and focussed.

The novelty is somewhat lacking in that their approach might be unique, but the outcomes are similar, if not identical, to other approaches.

**Strength And Weaknesses:**

This paper follows a clear logical flow moving from one concept to the next.

This paper is missing key vidualisation elements that make it difficult to interpret. For example, the image generation figures could have benefitted from a data visualisation technique that showed what was gained and generated universally between the different objects.

**Summary Of The Paper:**

This paper contributed a generalised slot-based approach for object centric representations as a Gaussian Mixture Model. To do so, this enables a set property prediction task. Finally, the representation of Gaussian Mixture Model can generalise to a variety of different object-orientated tasks. In their study they did note the gap of transfer learning not being extensively studied in this context.


**Summary Of The Review:**

Overall, the paper was well written, and although certain technical elements were missing from the paper, the overall quality is in line with the review.

---

> ### Author Response · Authors · 2022-11-17
> **Response to Reviewer koSa**
>
> We thank the reviewer for carefully reviewing our manuscript.
>
> **W1:** This paper is missing key vidualisation elements that make it difficult to interpret. For example, the image generation figures could have benefitted from a data visualisation technique that showed what was gained and generated universally between the different objects.
>
> - **Re:** We used the same visualization techniques as in SLATE's original paper: slot attention maps to specific image patches.

---

### Official Review · Reviewer_wLsE · 2022-10-26

**Confidence:** 4
**Correctness:** 3
**Technical Novelty And Significance:** 1
**Empirical Novelty And Significance:** 2
**Recommendation:** 3

**Clarity, Quality, Novelty And Reproducibility:**

# Clarity and Quality
Middle. The paper is not well-written, with some clear grammar errors, typography issues, and implausible references. For example,  in *abstract*: entities -> entity, in *intro*: centir -> centric. The writing is rough, and some sentences are long and not easy to understand. It seems the paper is not finished entirely.




**Strength And Weaknesses:**

# Strength
1. The SMM model integrates the learnable slots into the GMM model by replacing the clustering assignment and center update with learnable functions, such design learns the mixture model in an end-to-end manner.
2. The paper conducts some experiments to compare with other object-centric models, demonstrating its efficacy in terms of performance.

# Weakness
1. The SMM model learns to update the slots with a density function and distribute the learned density using a mechanism similar to slot competition. Such design is basically the composition of slot attention and GMM with incremental contribution, the novelty is not sound.
2. The experimental results are not sufficient to convince me that SMM is a promising model that has certain significant advantages over the slot-attention model or GMM. It achieves SOTA performances on set prediction, which is not sufficient for evaluating object-centric learning, direct metrics for object-centric learning such as ARI and PSNR should be used to evaluate.
3. Table 2 does not contain any comparison with other methods, same as Figure1 and Figure2.
4. There is no ablation or further analyses for the modules in SMM, makes its performance mistery and hard to explain.


**Summary Of The Paper:**

This paper proposes to combine the slot-based model with the gaussian mixture model (GMM) to improve the object-centric model. It explicitly represents the slot as the clustering center and uses the distance between slots to learn the mixture model. The experiments show certain improvements compared with some previous models.

**Summary Of The Review:**

Overall, I think the paper is not ready and the contributions are incremental, detailed justification are list in *weakness*. I suggest the author to submit it to next venue with sufficient experiments and analyses. The SMM model might deserve further investigation for its potentials in object-centric learning and broader domains.

---

> ### Author Response · Authors · 2022-11-17
> **Response to Reviewer wLsE**
>
> We thank the reviewer for carefully reviewing our manuscript.
>
> **W1:** The SMM model learns to update the slots with a density function and distribute the learned density using a mechanism similar to slot competition. Such design is basically the composition of slot attention and GMM with incremental contribution, the novelty is not sound.
>
> - **Re:** Object-centered learning based on slot attention is the focus of research. By now, several improvements to this mechanism have already been proposed, which we have listed in related works. It should be noted that many of them are incremental improvements from the point of view of algorithmic implementation, comprising one line of code. However, from the point of view of moving towards a more versatile approach and especially considering metrics on benchmarks, it is admitted that they all play an important role in the object-centric approach. Our idea is implemented quite simply, but seriously improves learning stability. Our approach also suggests looking at the slot competition process from a new point of view and proving that using SMM here gives results. SMM represents slots not only as centers of clusters, but also uses information about the distance between clusters and assigned vectors, which leads to more expressive slot representations.
>
> **W2:** The experimental results are not sufficient to convince me that SMM is a promising model that has certain significant advantages over the slot-attention model or GMM. It achieves SOTA performances on set prediction, which is not sufficient for evaluating object-centric learning, direct metrics for object-centric learning such as ARI and PSNR should be used to evaluate.
>
> - **Re:** We argue that AP in the set property prediction task is, to the same extent, a direct metrics for object-centric learning as ARI. The difference is that ARI is used to evaluating the quality of masks that are restored from objects. In our work, we focused on the growth in metrics for the set property prediction task on the CLEVR dataset because here Slot Attention does not demonstrate such good results in absolute values as in restoring objects with masks, i.e. the difference between the approaches is more clearly visible quantitatively (Slot Attention shows 98.8 ± 0.3 ARI on CLEVR6). But we also computed the ARI metric on the CLEVR10 dataset (we used 300k instead of 500k training steps) ​​ and showed results in Table 3.
> |Model | ARI|
> |--------|-----|
> |SA|85.9|
> |SMM|**91.3**|
>
> **W3:** Table 2 does not contain any comparison with other methods, same as Figure1 and Figure2.
>
> - **Re:** Figures 3, 4 and 5 as well as Table 2 compare our proposed model (Slot Mixture Module) with the original Slot Attention.
>
> **W4:** There is no ablation or further analyses for the modules in SMM, makes its performance mistery and hard to explain.
>
> - **Re:** We have added a comparison between vanilla K-Means and GMM clustering approaches in the set property prediction task that demonstrates prior advantage of the Mixture Model based approaches (Table 4).
> |Model|$AP_\inf$|$AP_1$|$AP_{0.5}$|$AP_{0.25}$|$AP_{0.125}$|
> |-------|-----------|---------|-------------|--------------|---------------|
> |K-Means|81.7|49.1|7.2|1.4|0.2|
> |GMM |**88.6**|**53.3**|**9.2**|**2.3**|**0.5**|
>
> **Comments on clarity and quality:**
> We carefully eliminated all spelling, grammatical and other errors in the text of the paper.

---

### Author Response · Authors · 2022-11-17
**General Response**

Thank you to all reviewers for taking the time and effort to read through our work and giving comments and feedback.

We posted a new version where we took considered all the main remarks (added a method scheme, explanations to the background, additional experiments etc.) and we ask everyone to briefly familiarize themselves with it. All main changes are highlighted in color for convenience.

We look forward to engaging the reviewers and are committed to address any further concerns.

---

### Decision · Program_Chairs · 2023-01-20

**Decision:**

Reject

**Justification For Why Not Higher Score:**

To meet the bar for acceptance and to justify the added complexity of this model over Slot Attention, the authors should consider providing stronger empirical evidence demonstrating the benefits of the model, going beyond simple synthetic tasks. The paper would also heavily benefit from a stronger focus on self-supervised tasks as opposed to supervised set prediction (for which Slot Attention is no longer a strong baseline) and general improvements in quality and clarity of writing / exposition.

**Justification For Why Not Lower Score:**

N/A

**Metareview: Summary, Strengths And Weaknesses:**

This paper addresses the problem of object-centric representation learning for images using a slot-based neural network. The paper presents a variant of the Slot Attention [1] mechanism that resembles a Gaussian mixture model (GMM) as opposed to a soft k-means clustering algorithm (used in the original Slot Attention mechanism). The method is validated on supervised object detection and property prediction (framed as set prediction), and unsupervised object discovery.

I agree with the reviewers that the investigation of the Slot Attention mechanism under a Gaussian mixture model perspective is interesting. While the paper generally shows quantitative improvements over the original Slot Attention model, these improvements are, however, primarily focused on the supervised set prediction setting of Slot Attention on a very simple task (object property prediction on CLEVR). The field of set-based object detection (and property prediction) has significantly advanced since Slot Attention was published, primarily with methods that build on top of Detection Transformers (DETR) [2]. While these results are promising (as pointed out by reviewer *LBbF*), an improvement over Slot Attention on a synthetic supervised task like the one investigated here is not sufficient to carry the weight of a full conference paper (as highlighted by, e.g., reviewer *oeon*). Other weaknesses of the paper pointed out by the reviewers include the insufficient clarity of writing, concerns about the claimed relation to GMMs (i.e. correctness of the algorithm), and the significance of the remaining experimental evaluation.

During the discussion period, the authors have added additional experiments (incl. on object discovery) and improved the writing, which alleviate some of these concerns. Overall, however, I agree with the reviewers that this paper does not yet meet the bar for acceptance.

[1] Locatello et al., Object-Centric Learning with Slot Attention (NeurIPS 2020)
[2] Carion et al., End-to-End Object Detection with Transformers (ECCV 2020)